

# THE IMPACT OF RISING ATMOSPHERIC $CO_2$ LEVELS AND RESULTING OCEAN ACIDIFICATION TO THE PHYSICAL (SOLUBILITY) OCEAN PUMP OF $CO_2$.

Wolfgang Dreybrodt
Faculty of Physics and Electrical Engineering, University of Bremen, Germany
dreybrodt@ifp.uni-bremen.de
**Abstract**
An alternative measure of the ocean's carbonate buffer system efficiency to absorb $CO_2$
from the atmosphere is proposed. Instead of the Revelle factor R = (ΔCO2/CO2)/(ΔDIC/DIC) =
(DIC/CO2)/ (ΔDIC/ΔCO2) the sensitivity S = (ΔDIC/ΔCO2) is preferable because it gives
directly the change ΔDIC of the concentration of DIC in the seawater caused by the change
ΔCO2 of carbon dioxide in the atmosphere. To this end the DIC concentration of seawater at
temperature T in equilibrium with a defined $CO_2$ level in the surrounding atmosphere is
calculated by use of the geochemical program PHREEQC.  From the function DIC(CO2,T) one
obtains by differentiation the sensitivity  S = dDIC/dCO2 =  ΔDIC/ΔCO2 and also the Revelle
factor R. Using S as the change of the ocean's buffer capacity reveals a better insight of its
future evolution than using the Revelle factor R.
One finds that the buffer capacity S has declined by about 30% from 1945 to present and
that its future decline from 400 to 600 ppm will be a further 30%.  By calculating the uptake
of $CO_2$ of his equilibrium pump an upper value of 1.3 Gigatons/year is obtained, small in
comparison to the 10 Gigatons/year absorbed by the ocean at present. The Revelle factor R
at present is calculated R = 13 and rises to 18 at a $CO_2$ level of 800 ppm. This increase of R



has been interpreted as indication of the collapse of the solubility pump. S and R, however,
are defined from equilibrium chemistry and are a measure of the $CO_2$ absorbed by the
ocean's upper mixed layer by increase of the $CO_2$ level in the atmosphere without regarding
its sinking into the deep-ocean by the thermohaline circulation. The difference ΔDIC
between the actual value and the value at 280 ppm is transported into the deep-ocean by
the global meridional conveyor belt.  ΔDIC increases with increasing $CO_2$ level. At 280 ppm
the system ocean-atmosphere is in equilibrium and the sink is zero. At 400 ppm a value of
about 1.9 Gtons/year is estimated that increases to 3.9 Gtons/year at 600 ppm and to 5
Gtons/year at 800 ppm. At present $CO_2$ level increase of 2ppm/year 10 Gtons/year are
absorbed by the ocean. The solubility pump contributes 3.2 Gtons/year: 1.3 Gtons/year by
equilibrium absorption into the mixed layer and 1.9 Gtons/yeat by thermohaline circulation.
At 600 ppm the total sink is 4.6 Gtons/year and at 800 ppm 5.5 Gtons/year. To conclude, the
solubility pump is not endangered by ocean acidification. In contrast, it increases with
increasing $CO_2$ level of the atmosphere to yield significant contribution.

**1. Introduction**
Only one half of anthropogenic $CO_2$ emitted remains in the atmosphere. About one quarter
is absorbed by the land sink via vegetation. The remaining quarter sinks into the ocean by
the biological pump and the physical (solubility pump) **(**Friedlingstein et al., 2022**).** The ocean
$CO_2$ sink has increased steadily with rising $CO_2$ level since the beginning of industrialisation.
As an example, $CO_2$ level of 317 ppm in 1960 raised to 420 ppm in 2021 and accordingly the
ocean sink from 1.1 ± 0.4 GtC/yr in 1960 to 2.8 ± 0.4 GtC/ y during 2021 (Friedlingstein et al.,
2022**).** Thus, the ocean sink has increased proportional to the rise in atmospheric $CO_2$.To
predict the future evolution of the $CO_2$-concentration (ppm) in the atmosphere by models



one has to know whether this increase will be permanent. One part of the oceanic sink is the
solubility pump that transports dissolved inorganic carbon (DIC) in equilibrium with the
partial pressure $p_{CO2}$ in the atmosphere (0.0001 atm ≙ 100 ppm) into the deep ocean. The
future effectivity of this physical pump has been questioned because with increasing
acidification of the ocean its buffering capacity decreases. This is commonly expressed by
the Revelle factor R (Zeebe and Wolf-Gladrow, 2001, Eglestone et al., 2010).
R = (ΔDIC/DIC)/(ΔCO2/CO2) = (ΔDIC/ΔCO2)/(DIC/CO2).
ΔDIC is the change in concentration DIC caused by a small increase ΔCO2 of the
concentration $CO_2$ in the atmosphere. $CO_2$ and DIC are the corresponding concentrations.
However, the Revelle factor is used mostly  only qualitatively stating that increasing values
of R indicate weakening of the buffer capacity (e.g., Climate Change 2007: The Physical
Science Basis. AR4 IPPC, Bates and Johnson, 2020). A more appropriate measure, the
sensitivity
S = ΔDIC/ΔCO2 has not been used in the scientific community. Middelburg et al., 2020 state:
"there are few studies where buffer and/or sensitivity factors are being used, except for the
well-known Revelle factor."  To judge quantitively the decrease of buffer capacity that gives
the amount of DIC increase by reaction of $CO_2$ to $HCO_3^-$ and $CO_3^-$ the evolution of sensitivity S
in dependence on the $CO_2$ level in the atmosphere is a better alternative. To this end I
calculate using the geochemical program PHREEQC (Parkhurst and Appelo, 2013) the
chemical composition of sea water in chemical equilibrium with $CO_2$ of defined partial
pressure $p_{CO2}$ (ppm ) in the surrounding atmosphere at defined temperature T. This way
DIC(pCO2,T) as a function of $p_{CO2}$ and T  is obtained. By differentiation one gets dDIC/dpCO2
= ΔDIC/ΔCO2 = S at defined temperature. From this I discuss the decrease of buffer capacity
with increasing $p_{CO2}$. I report the Revelle factor R as a function of S to enable quantitative



arguments using the Revelle factor R. This equilibrium pump does not consider the
overturning circulation of the ocean that transports the water of the mixed zone into deep-
ocean. This transport pump increases steadily with increasing $p_{CO2}$. The physical pump is the
sum of the sink by the equilibrium pump and the overturning transport pump. It increases
steadily to yield significant contributions.

**Methods**
The input file of the program PHREEQC is shown in Table 1. The first block `SOLUTION 1`
defines the composition of sea water including major elements and boron. The second block
EQUILIBRIUM_PHASES equilibrates this solution with gaseous $CO_2$ of the surrounding
atmosphere. Input parameters are temperature "temp" in °C and CO2(g) as log($p_{CO2}$) where
$p_{CO2}$ is in atm.
From the output file one can read pH and extract the concentrations of DIC (C(4))  and its

```
SOLUTION 1   Seawater
units ppm
pH 8.22
pe 8.451
density 1.023
temp 5
Ca 412.3
Mg 1291.8
Na 10768.0
K  399.1
Si 4.28
Cl 19353.0
B  4.5
Alkalinity 141.682
     as HCO3
S(6) 2712.0
EQUILIBRIUM_PHASES
CO2(ag)  -2.921
END
```

```
      pH = 7.715

 DIC C(4)       2.425e-03

HCO3-          1.753e-03
 MgHCO3+       3.129e-04
 NaHCO3        2.253e-04
 CO2           6.588e-05
 CaHCO3+       3.420e-05

   8.017e-06
 NaCO3 MgCO3
    1.542e-05
 CO3-2      -
    5.031e-06
 CaCO3         4.984e-06
```


**Table 1: Input file of PHREEQC**                **Table 2: Output results**
species in mol/kg. The program includes ion pairs with Ca and Mg.  $MgHCO_3^+$ and $NaHCO_3$



occur in considerable concentrations. In programs that do not include ion pairs these are
included as $HCO_3^-$. I have calculated DIC and pH from $CO_2$ levels of 300 ppm in steps of
33ppm up to 800 ppm. The data points were transferred to the program Origin. Then they
were fitted to a $5^{th}$ order polynomial (R-square = 0.99995; SD = $3,3\cdot10^{-4}$; $p<10^{-4}$) to smooth the
data for differentiation performed by the program. The figures were created by the graphics
of Origin.

**3. Results**
Fig.1 represents the results for DIC at fixed temperature T = 15°C (solid line). The dashed line
depicts the path where according to the increasing $CO_2$ level the temperature increases

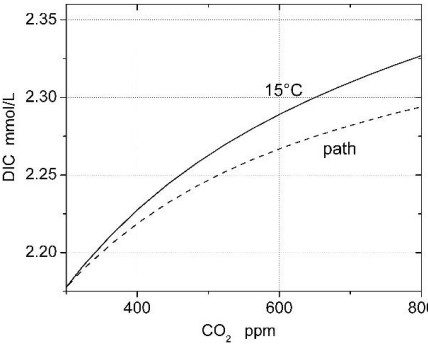

**Fig. 1: DIC at fixed temperature of T = 15°C (solid line). The dashed line depicts DIC when temperature changes with increasing $CO_2$**    **Fig. 2: Sensitivity S = dDIC/dCO2 for fixed temperature (solid line) and the path (dashed line) taking into account temperature increase with increasing $CO_2$ level. See text.**

linearly by 0.01 °C per 1ppm increase of $CO_2$ level corresponding to their linear correlation
obtained from NASA data of temperature and $CO_2$ level. The curve starts at T = 15°C, 300
ppm with steps of 33 ppm and 0.34°C and ends at 800 ppm and T = 19.1 C. Due to the rising
temperature DIC is reduced slightly  in comparison to fixed temperature a1 15 °C. Fig 2
depicts the sensitivity S = dDIC/dCO2 obtained from differentiation of the curves in Fig. 1.





dDIC is the change in the concentration of DIC in mmol/L that is caused by an increase of $CO_2$
by $dCO_2$ in ppm. This change can be converted as change of the aqueous $CO_2$ concentration
$c_{aq}$ in the liquid by Henry's law   $c_{aq} = K_H \cdot p_{CO2}$. At 15°C for sea water, $K_{H=}$ 0.04 mol/atm
(Zeebe and Wolf-Gladrow,2001).  For 1ppm the change $dc_{aq} = 4 \cdot 10^{-5}$ mmol $CO_2^{aq}$. The
corresponding change dDIC = S·1ppm = 0.0004 mmol DIC. Defining S*in units of
$mmolDIC/mmolCO_2^{aq}$ this way S*= 0.0004/0.00004 mmolDIC/mmol $CO_2^{aq}$ = 10

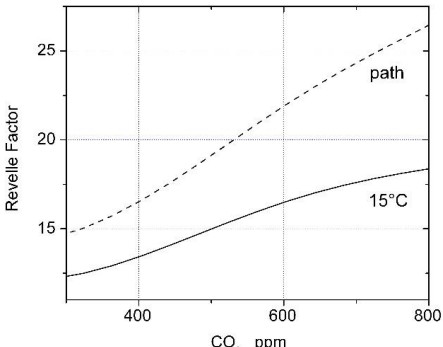

Fig. 3: Revelle factor for fixed temperature (solid line) and path (dashed line).

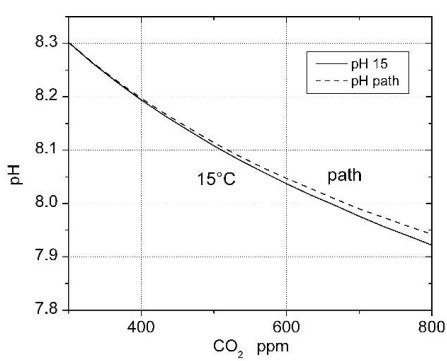

Fig. 4: pH for fixed temperature (solid line) and path (dashed line).


$mmolDIC/mmolCO_2^{aq}$ at 400 ppm. This means that 10 units $CO_2$ have been absorbed of
which 9 units have reacted to carbonates. For low pH < 4 where all DIC is in $CO_2^{aq}$, S* = 1.  At
15°C the value of S = 0.0001 corresponds  to value S* = 2.5. Note that this conversion
depends on temperature due to the temperature dependence of $K_H$ (0.051 at 5°C, 0.038 at
15°C. and 0.029 at 5°C). Both sensitivity curves show a steady drastic decline of the buffering
capacity from 0.0006 to 0.00016 (mmol/L)/ppm at fixed temperature and from 0.0005 to
0.00011 (mmol/L)/ppm for the path. Thus, the reduction by doubling $CO_2$ from 300 ppm to
600 ppm means a reduction to 42 % at fixed temperature and 33 % for the path.
The corresponding Revelle factors R = (DIC/CO2)/(dDIC/dCO2)  are shown in Fig. 3. They




illustrate why the Revelle factor cannot be used easily as quantitative measure because the
reduction of buffer capacity is by its change and not by its absolute value. Therefore, one has
to know the end values.  In contrast, sensitivity S gives the reduction from the known initial
value. In other words, the large background of R at 300 ppm prevents a reasonable
interpretation. Finally, in Fig. 4 acidification of ocean, the reason for declining buffer capacity
is shown as pH versus $CO_2$ level. pH drops almost linearly with $CO_2$ level from pH = 8.3 at 300
ppm to 7.9 at 800 ppm. There is little difference between constant temperature at 15°C and
the path regarding global warming. The change in pH is close to the projection of Jiang et al.,
2019 using the RCP 6.0 scenario of IPCC. This holds also for the change of the Revelle factor
R in Fig. 3.

**4. Discussion**
To obtain some overview on the variability of sensitivity S and Revelle factors R in Fig. 5 one
finds DIC for 5, 15, and 25°C respectively. The corresponding sensitivities S are shown in Fig.
6 and the Revelle factors are depicted in Fig. 7. For completion pH is illustrated in Fig. 8.

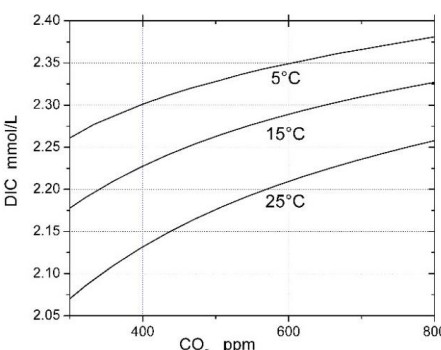

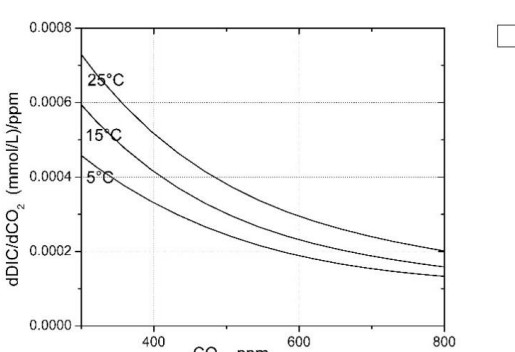

**Fig. 5: DIC concentration for various temperatures**

**Fig. 6: Sensitivity S = dDIC/dCO2 for various temperatures**



At fixed $CO_2$ the DIC concentration (Fig. 5) decreases with temperature whereas the
sensitivity S increases as can be seen from the slopes increasing with rising temperature.
These slopes are shown as S = dDIC/dCO2 in Fig. 6. S increases with rising temperature. As
one can read from Fig. 6 an increase of temperature by 5°C causes a reduction of the initial
value at 15°C by about 10% for all $CO_2$ levels. The impact of changing $CO_2$ level by far exceeds
that of increasing temperature.
S decreases with increasing $CO_2$ level. It is important to note that one finds a reduction by 36
% at the beginning from 300 to 420 ppm, corresponding to the time from 1945 to 2021.
Further reduction from 400 to 500 ppm is 16 % and continues to decrease further on for all
temperatures. This is in contrast to the opposite behaviour of the Revelle factor in Fig. 7.
One finds a small increase at the beginning up to 400 ppm followed by rise about twice of
the initial one for $CO_2$ between 400 to 600 ppm, valid for all temperatures. Thus, using S as
measure for impact to the oceans buffer capacity leads to conflictive conclusion about future

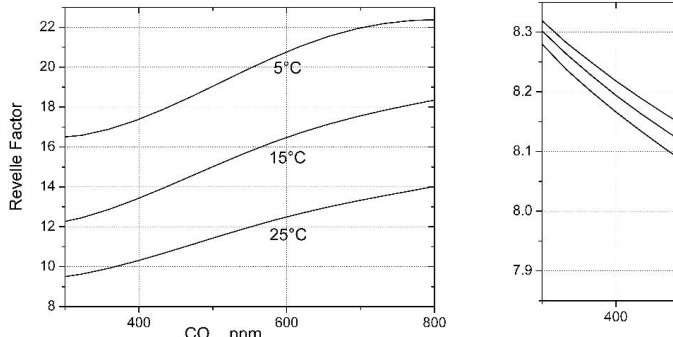

**Fig. 7: Revelle factor for various temperatures**    **Fig. 8: pH for various temperatures.**

evolution with consequences in defining pathways for $CO_2$ emissions in climate change
policy. At present public policy seems to be convinced that at least the physical ocean pump
will fail in the near future. Although the mixed layers capacity has been reduced by about
30% of its initial value for all temperatures during 1945 (300ppm) to 2015 (400ppm). The





ocean sink (Friedlingstein et al., 2022), however has continuously increased during this time
span. This leads to the conjecture that the physical sink into the mixed layer may not
contribute as significantly to the total ocean sink as thought by using the concept of
equilibrium chemistry (Revelle factor).
Finally, to relate sensitivity S to Revelle factor R, Fig. 9 illustrates R as a function of S.

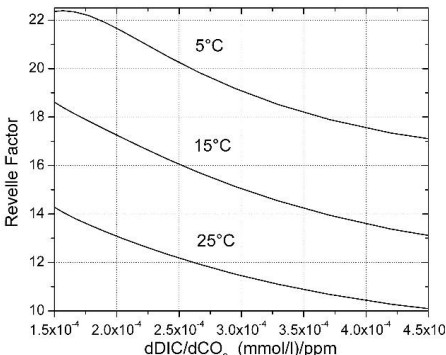

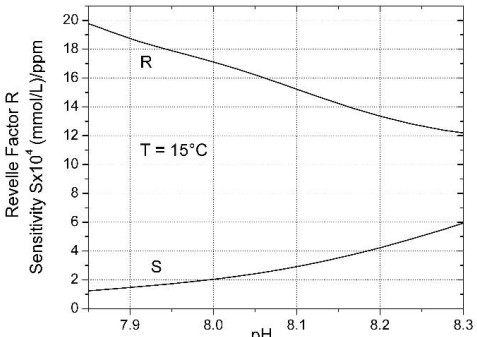

**Fig. 9: The Revelle factors R in relation to the sensitivity S for various temperatures**

**Fig. 10: The Revelle factor R and sensitivity S as function of pH at 15°C**

It is obvious why using S should be preferred. If S changes from $1.5 \cdot 10^{-4}$ to $4.5 \cdot 10^{-4}$ by 200 %
the corresponding chage in R is only about 40 % for 25°C and 20% for 5°C. Therefore, S gives
a more realistic view. Fig. 10 shows R and S as function of pH at 15°C. R changes from 12 to
20 with pH dcreasing from 8.3 to 7.85. But, in contrast to the sensitivity from its value no
direct meaning can be derived. From its defition a simple relation is: $R = 2.27/(CO2 \cdot S)$
because DIC ≈ 2.27 mmol/L remains constant within a few percent (see Fig. 1 and Fig. 5).
From this one may understand why R is used only qualititavely to judge ocean's physical
pump buffer capacity.
Using the DIC data one can estimate the upper limit of the present $CO_2$ flux from the
atmosphere to the ocean's mixed layer. I calculate the volume $V_1$ of the upper 1 meter of the
mixed layer $V_1 = 0.71 \cdot 4\pi R^2 \cdot 1\ m^3 = 3.6 \cdot 10^{17}$ L. R is Earth radius and 0.71 ocean coverage. The



amount $M_1$ of DIC that can be absorbed with a sensitivity $S_1 = 10^{-7}$ (mol/L)/ppm is $M_1 = V_1 \cdot S_1$
mol/ppm. Consequently, the amount $M_t$ absorbed by a mixed layer with depth t(m) and a
change of n ppm $CO_2$ is $M_t = M_1 \cdot t \cdot s \cdot n$ mol $CO_2$ when sensitivity $S = s \cdot S_1$. Converting to g $CO_2$
one has to multiply by the molecular weight 44 g/mol of $CO_2$ to obtain
$M_t = M_1 \cdot t \cdot s \cdot n$ mol$\cdot$44g/mol $= M_1 \cdot t \cdot s \cdot n \cdot 44$ (g).
A reasonable estimation of the mixed layer depth t is 100 m (de Boyer et al., 2004, Boyer et
al., 2022, Birol Kara et al., 2000, Doney et al., 2004). At present the increase of $CO_2$ is
2 ppm/year (n = 2) and (s = 4). Using these numbers, one finds $M_t$ = 1.3Gigatons/year. The
value of s = 4 corresponds to a temperature of 15°C at 420 ppm (see Fig. 6). This assumption
is reasonable because oceans temperature is distributed between 25° at the equator to 5°C
in the polar oceans.
Another argument must also be considered. Carbon is absorbed by the ocean where water
sinks to the deep ocean. At regions of upwelling water rich in $CO_2$, however, $CO_2$ is released
into the atmosphere (Landschützer et al., 2014, Crisp et al., 2022). This water outgasses $CO_2$
into an atmosphere with higher partial pressure. This causes a reduced flux of outgassing
and the difference of outgassing between higher and lower partial pressure at the intake
acts as effective influx in upwelling regions and justifies the assumption.
It must be stressed that the flux calculated so far by equilibrium chemistry represents the
capacity to absorb $CO_2$ from the atmosphere by a stagnant isolated mixed layer that does
not sink into depth. Therefore, this sink is caused by equilibrium chemistry and could be
termed as equilibrium sink (pump). This pump declines with increasing acidification of the
ocean. At pH < 4 the only existing carbonate species are aqueous $CO_2$ and $H_2CO_3$. Therefore,
the absorption of $CO_2$ is governed by Henry's law. Therefore, dDIC/dCO2 = $K_H$ and stays
constant with further decreasing pH. At 15°C dDIC/dCO2 = $4\cdot10^{-5}$ mmol/ppm. This



corresponds to a flux of 0.13 Gt/year, an almost total breakdown of the mixed layer's
capacity to absorb $CO_2$.
This, however, does not mean that the physical pump breaks down as has been concluded
from the increase of the Revelle factor. In IPCC AR4 one finds: "The ocean's capacity to
buffer increasing atmospheric $CO_2$ will decline in the future as ocean surface $pCO_2$ increases
(Figure 7.11a). This anticipated change is certain, with potentially severe consequences."
(Denman et al., 2007).
The total $CO_2$ sink consists of two parts: the equilibrium sink as already stated and the
transport sink. This is governed by the global meridional overturning circulation where
surface waters of the mixed layer flow from the equator to the polar regions and sink there
into the deep ocean by thermohaline circulation. In the North Atlantic deep water formation
is 15±2 Sv (1 Sv = $10^6$ $m^3$/s) and 21±6 Sv in the southern ocean (Ganachaud and Wunsch,
2000, Rahmstorf, 2002). These waters have cooled to low temperatures (about 5°C) when
they sink. They transfer the $CO_2$ in the mixed layer that contains also the anthropogenic
carbon into deep-ocean. These waters are replaced by upwelling waters back to the surface
without anthropogenic carbon (Terhaar et al., 2022) that readily absorb $CO_2$ from the
atmosphere until equilibrium is established.
Dividing the volume $V_{mix}$ of the mixed layer by the global formation of deep water of 36±6 Sv
one obtains, $\tau_{drain}$ , the time needed to drain that layer into the ocean as 57 years. The time
for chemical equilibration to a change of atmospheric $CO_2$ is on the order of 1 year (Jones et
al., 2014). Therefore, DIC in the mixed layer is in equilibrium with the $CO_2$ in the atmosphere
as given in Fig.5. At a $CO_2$ level of 280 ppm the flux of $CO_2$ into the ocean is zero and the
system is in equilibrium (Friedlingstein et al., 2022). With increasing $CO_2$ level, the deviation
of DIC equilibrium concentration is given by $(DIC_{ppm}- DIC_{280}) = \Delta DIC_{ocean}$.  $\Delta DIC_{ocean}$ represents



the amount of anthropogenic carbon absorbed into the mixed layer since the onset of
industrialisation. The flux $F_{ocean}$ into the ocean is given by $\Delta DIC \cdot V_{mix} / \tau_{drain} = F_{ocean}$ in
Gtons/year. At 400 ppm one finds a value of $F_{ocean}$ = 1.9 Gtons/year
Fig. 9 depicts $F_{ocean}$ in dependence on the $CO_2$ level. $F_{ocean}$ does not increase linearly with $CO_2$

level but increases with declining slope to 0.0087 Gtons/(year ppm) that stays constant for $p_{CO2}$ > 1 atm as calculated by PHREEQC. This way for an increase of 1 ppm/year of $CO_2$ level, $F_{ocean}$ increases by 0.0087 Gtons/year.

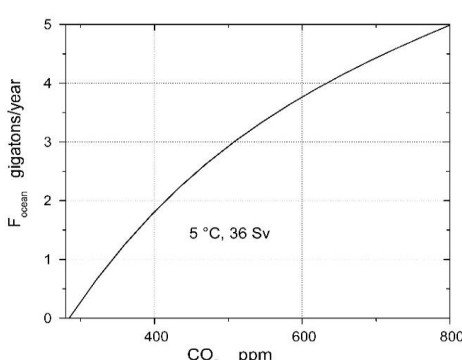

**Fig. 11: Flux $F_{ocean}$ into the ocean in dependence on $CO_2$ level**

At present the total Ocean sink is 10 Gt/year.  If at 400 ppm  a total sink of 3.2 Gt/year  is
correct the contribution of the physical pump is relatively small. It is possible that the
biological pump (Hauck and Völker, 2015, Riebesell et. al., 2007) compensates for this. In
view of the fact that this estimation might be critiqued it should motivate further research
and discussion in ongoing projects.

**5. Conclusion**
An alternative measure of the ocean's carbonate buffer to absorb $CO_2$ from the atmosphere
is proposed. Instead of the Revelle factor R = ($\Delta CO2/CO2$) /($\Delta DIC/DIC$) = (DIC/CO2)/
($\Delta DIC/\Delta CO2$) the sensitivity S = ($\Delta DIC/\Delta CO2$) is preferable because it gives directly the
change $\Delta DIC$ of the concentration of DIC in the seawater caused by the change $\Delta CO2$ of
carbon dioxide level in the atmosphere. To this end the DIC concentration of seawater in





equilibrium with a defined $CO_2$ level in the surrounding atmosphere is calculated by use of
the geochemical program PHREEQC.  From the function DIC(CO2) by derivation one obtains
the sensitivity S = dDIC/dCO2 =  ΔDIC/ΔCO2 and also the Revelle factor R.
Using S, the change of the ocean's buffer capacity better insight of its future evolution is
obtained than by use of the Revelle factor R.
S declines heavily since 1945 until it breaks down at $CO_2$ levels of 800ppm. One has to
consider, however, that R and S are calculated by equilibrium chemistry that does not
contain the sink caused by the thermohaline overturning circulation that transports the
water of the mixed zone into deep-ocean. S therefore, gives the amount of carbon as ΔDIC
that is stored in the mixed layer when the $CO_2$ level increases by ΔCO2.
The total solubility sink consists of two mechanisms: The equilibrium pump as described and
the transport pump that is caused by the global meridional overturning  circulation of 36 Sv.
This transfers into deep-ocean the difference $(DIC_{ppm}- DIC_{280}) = ΔDIC_{ocean}$ that has been
accumulated in the mixed layer from onset of industrialisation to the actual $CO_2$ level.
This sink increases continuously replacing the failure of the equilibrium pump. At 400 ppm
the total sink is 1.9, at 600 ppm it is 3.8 and at 800 ppm it amounts to 5 Gtons/year
depending solely on the $CO_2$ level in the atmosphere for ppm > 600.
To conclude, the total solubility pump is not endangered by ocean acidification. In contrast,
it increases with increasing $CO_2$ level of the atmosphere to yield significant contribution to
remove anthropogenic $CO_2$ from the atmosphere into deep-ocean.

**I declare that I do not have any competing interests.**

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
