# Peer review of "THE IMPACT OF RISING ATMOSPHERIC CO2 LEVELS AND RESULTING OCEAN ACIDIFICATION"

_EGUsphere, 2023_

## Referee Comment (RC1)

The author proposes a new factor called sensitivity given by
$$S = \Delta DIC/\Delta CO2$$
to measure of the ocean's carbonate buffer system efficiency to absorb $CO_2$ from the atmosphere. The S factor is a simplification of the Revelle factor by removing the $DIC/CO_2$ term.

This simplification does not allow to correctly apprehend the buffer capacity of seawater, as detailed below by two examples.

The efficiency of the ocean to absorb $CO_2$ from the atmosphere depends on the equilibrium in seawater of dissolved $CO_2$ and $HCO_3^-$ and $CO_3^{2-}$. This equilibrium depends on the actual concentrations of $HCO_3^-$ and $CO_3^{2-}$ and $CO_2$.

The higher the concentrations of $HCO_3^-$ and $CO_3^{2-}$, the "more" $CO_2$ can be dissolved (at equilibrium following Henry's law). This has been coined the ocean buffer capacity and was formalized by Revelle in his famous factor.

A quick example is given in Table 1.
For very different Total alkalinity (TA) values (300, 1300 and 2300), and for an increase of 1 µM of DIC the corresponding increase of CO2 is much higher in low buffered waters (low TA) than in the high buffered waters (high TA).

The proposed S factor, thus, is extremely variable as a function of TA variations in the surface of the ocean.

The higher the concentration of $CO_2$ already dissolved, the "less" $CO_2$ can be dissolved (at equilibrium following Henry's law). This is the other facet of ocean buffer capacity, as formalized by Revelle, and it was on this basis that he predicted that the capacity of the ocean to absorb $CO_2$ would slowly decrease in time.

This is illustrated by another quick example given in Table2. In this case the TA alkalinity is constant but the $CO_2$ concentration is variable.

The proposed S factor, thus, is extremely variable as a function of initial $CO_2$ concentrations in the surface of the ocean.

These two examples show that the proposed S factor is not useful because it does not account for the variations in $HCO_3^-$ (as given by TA for a constant initial $pCO_2$) and by the variations of $CO_2$ (as given by a constant TA for a variable initial $pCO_2$). On the contrary, this is accounted by the $DIC/CO_2$ term in Revelle's factor, and it does not make sense to simply the Revelle factor by removing this term.

**Table 1**

| Salinity (psu) | Temperature (°C) | TA (µmol/kg-SW) | DIC (µmol/kg-SW) | pH | pCO2 (µatm) | CO2 (µmol/kg-SW) | ΔCO2:ΔDIC (µmol:µmol) |
|---|---|---|---|---|---|---|---|
| 35 | 15 | 2300 | 2080.9 | 8.045 | 400.0 | 14.93 | |
| 35 | 15 | 2300 | 2081.9 | 8.043 | 402.2 | 15.01 | 0.08 |
| 35 | 15 | 1300 | 1210.0 | 7.821 | 400.0 | 14.93 | |
| 35 | 15 | 1300 | 1211.0 | 7.817 | 404.1 | 15.08 | 0.15 |
| 35 | 15 | 300 | 298.6 | 7.212 | 400.0 | 14.93 | |
| 35 | 15 | 300 | 299.6 | 7.199 | 413.6 | 15.44 | 0.51 |

**Table 2**

| Salinity (psu) | Temperature (°C) | TA (µmol/kg-SW) | DIC (µmol/kg-SW) | pH | pCO2 (µatm) | CO2 (µmol/kg-SW) | ΔCO2:ΔDIC (µmol:µmol) |
|---|---|---|---|---|---|---|---|
| 35 | 15 | 2300 | 2242.2 | 8.884 | 200.0 | 7.46 | |
| 35 | 15 | 2300 | 2243.2 | 8.877 | 203.1 | 7.58 | 0.11 |
| 35 | 15 | 2300 | 2280.5 | 8.594 | 400.0 | 14.93 | |
| 35 | 15 | 2300 | 2281.5 | 8.586 | 408.5 | 15.25 | 0.32 |
| 35 | 15 | 2300 | 2299.0 | 8.423 | 600.0 | 22.39 | |
| 35 | 15 | 2300 | 2300.0 | 8.413 | 613.5 | 22.90 | 0.50 |

Computations were made with CO2sys implemented in Excel using the carbonic acid dissociations constants of Mehrbach et al. on the pH Total scale.

---

## Author Comment (AC2)

Foreword

Both reviewers have not fully perceived the message of my work. Therefore, first I give a short overview of the basics of my paper. I wanted to provide a better understanding of the basic operation of the physical pump. This pump works in two steps. First, the carbonate buffer system of the upmost mixed layer reacts with the $CO_2$ of the atmosphere to attain chemical equilibrium. The question is:  what is the increase of DIC in the mixed layer when the partial pressure of $CO_2$ in ppm increases by $\Delta CO_2$. To this end I have calculated DIC (mmol/kg) as a function of $CO_2$ (ppm) using the program PHREEQC. Differentiating with respect to $CO_2$ gives $dDIC/dCO_2$ (mmol/kg ppm) termed as sensitivity S as a function of $CO_2$. S is the increase of DIC by increase of $CO_2$. It tells how much $CO_2$ is absorbed by the buffer system. S decreases with increasing $CO_2$. Usually, this buffering is described by the Revelle factor R. Therefore, I had to give the relation between S and R. Although, this is a side result it requires some text. Both reviewers have focussed to this part of the paper. The essential second part of the pump is transport of water of the mixed layer with high DIC by thermo-haline circulation into deep ocean and replacement by water in equilibrium with preindustrial $CO_2$ level. This part of the pump increases steadily with increasing $CO_2$. I admit that this is a simple model that needs only the well-known constants of equilibrium chemistry provided by PHREEQC and the amount of waterflow into deep ocean in Sv. The result gives at least the correct order of magnitude of the observed $CO_2$ uptake from the atmosphere into the ocean by the physical pump. In summary, my model reveals the basics that may be hidden in many complex models that are not intelligible by non- specialists. This opens understanding to a larger part of the scientific community and to my knowledge has not been published before.

Reply to RC2:

In the following the arguments of the reviewer are in *italic*, my response is in normal, and parts copied from my paper are underlined.

There is some misunderstanding in the definition of the Revelle factor. The reviewer's definition is
R = (DIC/[CO2]) / (ΔDIC/ΔCO2) = (Δ[CO2]/[CO2]) / (ΔDIC/DIC) where [CO2] is the aquatic $CO_2$ concentration; both DIC and [CO2] are measured in gravimetric units (mol kg−1).
My definition is R = (ΔDIC/DIC)/(ΔCO2/CO2) = (ΔDIC/ΔCO2)/(DIC/CO2).  ΔDIC is the change in concentration DIC caused by a small increase ΔCO2(gas) of the concentration $CO_2$ in the atmosphere. ΔDIC and DIC is in mol/kg and) ΔCO2 and CO2 is in ppm, the partial pressure of CO2(gas) in the atmosphere. Since the aquatic $CO_2$ concentration CO2(aq) is related to the partial pressure CO2(gas) in the atmosphere by Henry's law CO2(aq) = KH·CO2(gas) and ΔCO2(aq) = KH·ΔCO2(gas) both definitions are identical.

By rearranging one gets R = (Δ[CO2]/[CO2]) / (ΔDIC/DIC) = (DIC/CO2)/(ΔDIC/(ΔCO2) = (DIC/CO2)/S for both definitions and switching units in the lengthy comment is not necessary.

In line 160 ff in my work on finds: From its definition a simple relation is: R = 2.27/(CO2·S)  because DIC ≈ 2.27 mmol/L remains constant within a few percent (see Fig. 1 and Fig. 5).

After this lengthy discussion of relations between R and S, RC2 writes:

*"This is a misunderstanding, because with increasing atmospheric CO2 the uptake of CO2 by the ocean and its transport to deeper layers (solubility pump) will further increase. This is consistent with a decrease of the buffer capacity of the ocean with respect to increasing atmospheric CO2 which can be expressed by an increase of the Revelle factor or a decrease of the sensitivity (inversely related to each other). Based on this misunderstanding, the author addresses a problem that does not exist. I can not support publication of this paper."*

To resolve this misunderstanding I will add in the revision: R = (ΔDIC/DIC)/(ΔCO2/CO2) = (ΔDIC/ΔCO2)/(DIC/CO2). ΔDIC (mol/L) is the change in concentration DIC (moi/L) caused by a small increase ΔCO2 (ppm) of the concentration CO2 (ppm) in the atmosphere. CO2 can be also given as the concentration of aqueous CO2 in mol/L because CO2(atm) and CO2(aq) are related by Henry's law; CO2(aq) = KH·CO2(atm).

What is the misunderstanding? Such general remarks are not helpful.

Evidently, the reviewer has missed the second part of my paper where transport into the ocean is discussed. See Fig. 11 and Fig.5 and lines 249-257.

The total solubility sink consists of two mechanisms: The equilibrium pump as described and  the transport pump that is caused by the global meridional overturning circulation of 36 Sv.

This transfers into deep-ocean the difference (DIC ppm - DIC 280 ) = ΔDIC ocean  that has been  accumulated in the mixed layer from onset of industrialisation to the actual CO 2  level.

This sink increases continuously replacing the failure of the quilibrium pump. At 400 ppm  the total sink is 1.9, at 600 ppm it is 3.8 and at 800 ppm it amounts to 5 Gtons/year  depending solely on the CO 2  level in the atmosphere. For ppm > 600. To conclude, the total solubility pump is not endangered by ocean acidification. In contrast, it increases with increasing CO 2  level of the atmosphere to yield significant contribution to  remove anthropogenic CO 2  from the atmosphere into deep-ocean.

Detailed remarks to RC2:

p.2 *'At 400 ppm a value of about 1.9 Gtons/year is estimated that increases to 3.9 Gtons/year at 600 ppm and to 5 Gtons/year at 800 ppm.' Sentence needs more explanation ... 1.9 Gt C oceanic net uptake; estimated in current paper or from literature?)*

I do not understand this. Evidently this is estimated in the current paper.

*"Units: Gt C or Gt CO2? I guess always Gt C"*

units are in Gt CO2. See line 169-170. The numbers are derived in the current paper, see Fig. 11. I will use Gt $CO_2$ instead of Gt throughout the text, (mol/(Lppm)

*p.4 The input and output table of PHREEQC should be replaced by a proper list of relevant quantities with appropriate units (for example: what is meant by 'CO2(ag) - 2.921') The sensitivity as a function of DIC, TA, temperature, and salinity is easy to calculate with freely available software packages (compare, for example, Orr et al., 2015, Orr & Epitalon, 2015, Humphreys, et al, 2022, especially with CO2SYS available in MATLAB or Python on GitHub: https://github.com/jamesorr/CO2SYS-MATLAB, https://github.com/mvdh7/PyCO2SYS).*

I used the input and output files of PREEQC. All units are explained in the text. Maybe other programmes do the same job. But with the information given the reader can do calculations By PHREEQC.

In the following I comment some of the special remarks.

*p.6 'which 9 units have reacted to carbonates. For low pH < 4 where all DIC is in CO2aq, S\* = 1. At 15°C the value of S = 0.0001 corresponds to value S\* = 2.5. ' S, S\*: units missing.*

From 10 CO2 molecules absorbed 9 are converted to carbonates. Units of S are defined in the text.

*p.10 'Therefore, dDIC/dCO2 = KH' is wrong!*

The statement is correct only for pH < 4 when only CO2(aq) is existing. It was discussed to provide the limit at low pH.

*p.9 'From this one may understand why R is used only qualitatiavely [TYPO] to judge ocean's physical pump buffer capacity.' ???*

There are many comments with ????? but their question remains open.

I am helpless how to react to this review. I am afraid that the reviewer had only a restricted perception of the paper to provide a constructive review.

**In conclusion the review is highly biased to decline the paper and most of the objections are unfounded or even wrong. It does not give any hints how to improve the paper. Therefore, I cannot suggest any changes to the paper at present. I leave it to the editor how to proceed.**

---

## Author Comment (AC3)

Foreword

Both reviewers have not fully perceived the message of my work. Therefore, first I give a short overview of the basics of my paper. I wanted to provide a better understanding of the basic operation of the physical pump. This pump works in two steps. First, the carbonate buffer system of the upmost mixed layer reacts with the CO2 of the atmosphere to attain chemical equilibrium. The question is: what is the increase of DIC in the mixed layer when the partial pressure of CO2 in ppm increases by ΔCO2. To this end I have calculated DIC (mmol/kg) as a function of CO2 (ppm) using the program PHREEQC. Differentiating with respect to CO2 gives dDIC/dCO2 (mmol/kg ppm) termed as sensitivity S as a function of CO2. S is the increase of DIC by increase of CO2. It tells how much CO2 is absorbed by the buffer system. S decreases with increasing CO2. Usually, this buffering is described by the Revelle factor R. Therefore, I had to give the relation between S and R. Although, this is a side result it requires some text. Both reviewers have focussed to this part of the paper. The essential second part of the pump is transport of water of the mixed layer with high DIC by thermo-haline circulation into deep ocean and replacement by water in equilibrium with preindustrial CO2 level. This part of the pump increases steadily with increasing CO2. I admit that this is a simple model that needs only the well-known constants of equilibrium chemistry provided by PHREEQC, the depth of the mxed layer,i and the amount of waterflow into deep ocean in Sv. The result gives at least the correct order of magnitude of the observed CO2 uptake from the atmosphere into the ocean by the physical pump. In summary, my model reveals the basics that may be hidden in many complex models that are not intelligible by non- specialists. This opens understanding to a larger part of the scientific community and to my knowledge has not been published before.

 Reply to RC3:

In the following the arguments of the reviewer are in *italic*, my response is in normal, and parts copied from my paper are underlined.

     Reviewer 3 declines publication as follows:

*Prof Wolfgang Dreybrodt's presents a couple of different thoughts and back-of-the-envolope calculations on the ocean carbon sink. The main point of the manuscript is about simply rearranging the equation of the Revelle factor: R = (ΔDIC/DIC)/(ΔpCO2/pCO2) ⇔ ΔDIC/ΔpCO2 = R \* DIC/pCO2 = S.  As such, I do not believe that the scientific novelty or significance of this manuscript is worthy for publication.*

The definition of *R* = (ΔDIC/DIC)/(ΔpCO2/pCO2) is wrong. It must read
 (ΔDIC/DIC)/(ΔpCO2/pCO2) = 1/R and correspondingly *ΔDIC/ΔpCO2 = S = (DIC/pCO2)/R.*

ΔDIC/ ΔCO2 = S is only used in the paper to give a better understanding to the non-specialised community, how acidification weakens the buffer system because R is not easy to interpret. It is by no means the result of the paper.

The reviewer has not perceived the main message of my paper as explained above and to decline publication is unfounded.

My remarks to some of the reviewer's concerns to the ms.

*"In addition to the scientific novelty and significance, I have several major concerns regarding the manuscript."*

*"While it is recognised that the sink decreases with increasing Revelle factor (Revelle and Suess, 1957), even the high-emission scenario RCP8.5 is not estimated to lead to a collapse of the solubility pump (Rodgers et al., 2020)."*

This is exactly the result of my paper. See Fig. 11. The equilibrium pump becomes weak. The transport pump, however, increases. See lines 249-258 in the ms.

The total solubility sink consists of two mechanisms: The equilibrium pump as described and  the transport pump that is caused by the global meridional overturning  circulation of 36 Sv.  This transfers into deep-ocean the difference (DIC ppm - DIC 280 ) = ΔDIC ocean  that has been  accumulated in the mixed layer from onset of industrialisation to the actual $CO_2$  level.   This sink increases continuously replacing the failure of the equilibrium pump. At 400 ppm  3the total sink is 1.9, at 600 ppm it is 3.8 and at 800 ppm it amounts to 5 Gtons/year  depending solely on the $CO_2$  level in the atmosphere for ppm > 600.   To conclude, the total solubility pump is not endangered by ocean acidification. In contrast, it increases with increasing $CO_2$  level of the atmosphere to yield significant contribution to  remove anthropogenic $CO_2$  from the atmosphere into deep-ocean provided the thermo-haline circulation remains constant and is not weakened by climatic change.

Furthermore, the paper of Revelle, R., & Suess, H. E. (1957). Carbon dioxide exchange between atmosphere and ocean and the question of an increase of atmospheric $CO_2$ during the past decades. *Tellus*, 9, 1– 10 does not contain a definition of the Revelle factor although cited many times by copy and paste.

The Revelle factor has been defined by W. S. Broecker, T. Takahashi, H. J. Simpson, T.-H. Peng. Fate of Fossil Fuel Carbon Dioxide and the Global Carbon Budget. Science, 1979, Volume 206, Number 4417

*"The steps in pCO2 and T are far too large to calculate meaningful derivatives. It remains unclear to me why the author does not just use incremental steps or even better, directly the above-shown equation. This leads to several major errors throughout the*

*manuscript: In Fig. 2, for example, the sensitivity for 'path' and '15°C' at 300 ppm is different although it should be identical as T is still at 15°C for path."*

I have fitted the DIC data to a fifth order polynomial with high precision. From this the derivative with respect to CO2 is calculated to high precision. This does not lead to *"several major errors throughout the manuscript."* The reviewer should explain the major errors in some detail.
 Sensitivity is given by the slope of DIC(CO2), see Fig. 1 and Fig. 2 and therefore S is different for the path and 15°C at 300 ppm.
**In contrast, serious errors arise by following the reasoning of the reviewer.**

*"In Fig. 3, the Revelle factor should be identical for 'path' and '15°C'. Why is there such a large difference?"*

The Revelle factor is related to S by **R = DIC/(CO2·S) = 2.27/(CO2·S)** See lines 160-163 in the ms. As path and 15°C have different S, R must be different as well.
Lines 155-163 explain in detail the relation between R and S. This is copied from the ms in the lines that follow (underlined):
Finally, to relate sensitivity S to Revelle factor R, Fig. 9 illustrates R as a function of S.  It i obvious why using S should be preferred. If S changes from $1.5·10^{-4}$ to $4.5·10^{-4}$ by 200 %  the corresponding change in R is only about 40 % for 25°C and 20% for 5°C.
Therefore, S gives  a more realistic view. Fig. 10 shows R and S as function of pH at 15°C. R changes from 12 to 20 with pH decreasing from 8.3 to 7.85. But, in contrast to the sensitivity from its value no direct meaning can be derived. From its definition a simple relation is: R = 2.27/(CO2·S) because DIC ≈ 2.27 mmol/L remains constant within a few percent (see Fig. 1 and Fig. 5).  From this one may understand why R is used only qualitatively to judge ocean's physical pump buffer capacity.

*Units are not used in a careful manner. As an example: It is often not clear if it is Gt C or Gt CO2.*

As can be seen from lines 168 – 171 in the ms :" Consequently, the amount M t absorbed by a mixed layer with depth t(m) and a  change of n ppm CO 2  is M t  = M 1 ·t·s·n mol CO 2  when sensitivity S = s·S 1 .  **Converting to g CO 2   one has to multiply by the molecular weight 44 g/mol of CO 2  to obtain   M t  = M 1 ·t·s·n mol·44g/mol = M 1 ·t·s·n· 44 (g).**"

Therefore, where not stated otherwise the units are GtCO2 throughout the ms.

*There are many publications on the importance of the biological and solubility pump but there is no doubt that the solubility pump is the major contribution to the anthropogenic carbon sink (Friedlingstein et al., 2022). This example demonstrates why these 'back-of-*

*the-envelope' equations cannot be used when treating such a complex system as the global ocean.*

It would have been helpful if the reviewer had given a copy of this statement in a paper with 90 pages, thar the "*solubility pump is the major contribution to the anthropogenic carbon sink (Friedlingstein et al., 2022)."* I could not find it. (Friedlingstein et al., 2022) deal with the ocean sink and do not give the single contributions of the solubility and the biological sink but only the ocean sink that is the sum of both.

*Some assumptions are unreasonable. The author speaks about a time when pH decreases below 4. Even under the high-emission scenarios, such a low pH is not possible on average at the ocean surface.*

I do not speak "*about a time when pH decreases below 4."* Of course, pH = 4 is not reached in any scenario. The intent of this passage was to show the lower limit of the equilibrium sink. The reviewer evidently did not understand the meaning of this part of my paper, lines 184-189:

It must be stressed that the flux calculated so far by equilibrium chemistry represents the capacity to absorb $CO_2$ from the atmosphere by a stagnant isolated mixed layer that does not sink into depth. Therefore, this sink is caused by equilibrium chemistry and could be termed as equilibrium sink (pump). This pump declines with increasing acidification of the ocean. At pH < 4 the only existing carbonate species are aqueous $CO_2$ and $H_2CO_3$. Therefore, the absorption of $CO_2$ is governed by Henry's law. Therefore, $dDIC/dCO_2 = K_H$ and stays constant with further decreasing pH.

In conclusion the reviewer states:

"*Overall, I believe that this manuscript presents no new findings or results and the simplification of complex mechanisms, which are already presented in detail by Sarmiento and Gruber (2006), by simple equations with strong assumptions, leads to erroneous conclusions. Relatively simple 3-D biogeochemical models in the 1990s were already able to estimate the different parts of the ocean carbon sink in a much better and accurate way (Joos et al., 1999)."*

The reviewer does not tell what are the erroneous conclusions and how did they arise. My model uses equilibrium chemistry of the seawater carbonate system and the amount of water transported to deep ocean. It is possible to use it to predict the evolution of the physical pump upon impact of weakening of the thermo-haline circulation. The reviewer misses to show where his statement "*Relatively simple 3-D biogeochemical models in the 1990s were already able to estimate the different parts of the ocean carbon sink in a much better and accurate way (Joos et al., 1999)"* is given in this cited paper. Its abstract states: "A low-order physical-biogeochemical climate model

was used to project atmospheric carbon dioxide and global warming for scenarios developed by the 3 Intergovernmental Panel on Climate Change." There is no discussion about *the different parts of the ocean carbon sink* in that paper.

**In conclusion the review is highly biased to decline the paper and most of the objections are unfounded or even wrong. It does not give any hints how to improve the paper. Therefore, I cannot suggest any changes to the paper at present. I leave it to the editor how to proceed.**